# Expression of miR-34a-5p is up-regulated in human colorectal cancer and correlates with survival and clock gene *PER2* expression

**Kristina Hasakova[1], Richard Reis[2], Marian Vician[3], Michal Zeman[1], Iveta Herichova[1]***

**1** Department of Animal Physiology and Ethology, Faculty of Natural Sciences, Comenius University, Bratislava, Slovak Republic, **2** First Surgery Department, University Hospital, Comenius University Bratislava, Bratislava, Slovak Republic, **3** Fourth Surgery Department, University Hospital, Comenius University Bratislava, Bratislava, Slovak Republic

* herichova1@uniba.sk

**Data Availability Statement:** All relevant data are within the manuscript and its Supporting Information files.

## Abstract

Colorectal cancer represents a leading cause of cancer death. MicroRNAs (miRNAs) are small non-coding RNA molecules that have been extensively studied in tumours, since changes in their levels can reveal patient prognosis. Cancer progression is also influenced by the circadian system whose functioning is based on the rhythmic expression of clock genes. Therefore, we performed macroarray screening of tumour and adjacent tissues in patients undergoing surgery for colorectal carcinoma. We identified 17 miRNAs showing expression that was more than 100 times higher in tumour tissue compared to adjacent tissue. From *in silico* analysis, miR-34a-5p was selected as showing a computer-predicted interaction with *PER2*. Real-time PCR revealed a negative correlation between expression of *PER2* mRNA and miR-34a in patients with more advanced cancer stage. Expression of miR-34a was up-regulated in cancer tissue compared to adjacent tissue. High miR-34a expression was associated with better survival of patients. miR-34a showed lower expression levels in male patients with lymph node involvement, and a trend towards decreased expression in male patients with distant metastases. Male patients, but not female patients, with high expression of miR-34a and who were free of distant metastases and/or lymph node involvement showed better survival. Therefore, we proposed that expression of miR-34a was regulated in a sex-dependent manner and could be considered a marker of prognosis in earlier cancer stages in male patients.

## Introduction

Colorectal cancer (CRC) is the third most commonly diagnosed type of cancer, and the fourth leading cause of cancer deaths worldwide [1]. Since colorectal tumour development is slow and asymptomatic, it can progress undetected, increasing the importance of early detection as the most effective way to reduce mortality. In recent years, altered levels of microRNAs (miRNA) in tumour tissue compared to healthy surrounding tissue and plasma have been

**Funding:** This work was supported by APVV-14-0318, The Slovak Research and Development Agency, https://www.apvv.sk/?lang=en, to MZ; APVV-16-0209, The Slovak Research and Development Agency, https://www.apvv.sk/?lang=en, to IH; VEGA 1/0679/19, Scientific Grant Agency of the Ministry of Education, Science, Research and Sport of the Slovak Republic, https://www.minedu.sk/vedecka-grantova-agentura-msvvas-sr-a-sav-vega/, to IH. The funders had no role in study design, data collection and analysis, decision to publish, or preparation of the manuscript.

**Competing interests:** The authors have declared that no competing interests exist.

reported, making miRNAs potentially useful molecules for the screening of malformations, including CRC [2, 3].

miRNAs are short non-coding RNAs that regulate gene expression by translation repression and/or degradation of mRNA by binding to the 3' untranslated regions of specific target genes [4]. It has been estimated that over 60% of human protein-coding genes are regulated by miRNAs [5]. Many genes involved in cellular processes such as proliferation, differentiation and apoptosis are targeted by miRNAs. Depending on the function of the target genes, miRNAs can exert either a tumour suppressing or tumour promoting influence. One of the first studies that identified altered levels of miRNAs in CRC was conducted by Michael et al. [6], who detected a lower expression of miR-143 and miR-145 in tumour tissue compared to adjacent tissue. Later, it was shown that up-regulation of miRNA expression in colon cancer tissue compared to healthy tissue was more frequent than its down-regulation [2], e.g., the up-regulation of miR-21, which targets the tumour suppressor genes *programmed cell death protein 4* (*PDCD4*) and *phosphatase and tensin homologue* (*PTEN*) [7–9]. Since the pioneering studies in this field, the range of deregulated miRNAs in CRC has increased. In tumours of colorectal carcinoma up-regulated miRNAs include miR-17, miR-155, miR-146, miR-221, miR-31, miR196, miR-20a, miR92a, miR-130b, miR181b, miR-183, miR-196, miR-203, miR-215, miR-224 and miR-25, while down-regulated miRNAs include miR-10, miR-30, miR-22, miR-93, miR-126, miR-132, miR-139, miR-15, miR-195, miR340 and miR-106a [3, 10, 11].

The circadian system is an endogenous time-sensing system that enables organism to anticipate changes in the environment and adapt to them. Circadian rhythms refer to biological rhythms with a period close to 24 h, such as sleep/wake cycles, changes in hormonal levels or body temperature rhythm [12]. The circadian system is composed of the master oscillator situated in the *suprachiasmatic nucleus* (SCN) and peripheral oscillators located in most other tissues and cells of the mammalian body. Circadian rhythms are generated by a molecular transcriptional-translational feedback loop that consists of clock genes. Heterodimers composed of the transcription factors Brain and Muscle ARNT-Like 1 (BMAL1) and Circadian Locomotor Output Cycles Kaput (CLOCK) induce expression of the core clock genes *period 1–3* (*PER 1–3*) and *cryptochrome 1–2* (*CRY 1–2*). When a complex containing PER:CRY is translocated to the nucleus, the protein products of the *PER2* and *CRY* genes repress transcription of their own mRNAs by blocking BMAL1-CLOCK-mediated activity. After releasing PER:CRY complex inhibition, the basic feedback loop is closed. The whole process from induction of clock gene expression to release of PER:CRY repression takes approximately 24 h [13].

Disruption of the circadian system by modern lifestyles is becoming increasingly common, and growing evidences suggests that such changes may increase susceptibility to cancer in humans. Several epidemiological studies have revealed a positive association between shift-work and breast, colorectal and prostate cancer [14–16]. Other evidence is provided by experiments with rodents that show the acceleration of tumour growth by induction of SCN lesions or chronic jet lag [17].

In previous studies, we identified altered mRNA levels of the clock genes *PER2* and *CRY2* in colorectal carcinoma tissue compared to healthy tissue. The expression of *PER2* was significantly down-regulated in tumour tissue compared to adjacent tissue [18]. The tumour suppressive properties of *PER2* have been previously demonstrated. Mutant mice with a deletion in the PAS-B domain of *PER2* have increased sensitivity to γ-irradiation, with higher rates of malignant lymphomas compared to wild-type animals [19]. The down-regulation of *PER2* in CRC cells increased protein levels of β-catenin and cyclin D, which accelerated tumour cell growth [20]. On the other hand, overexpression of *PER2* in the mouse Lewis lung carcinoma cell line and mammary carcinoma cell line induced apoptosis connected with higher mRNA and protein expression of *TP53* and the apoptotic protein *Bax*, and lower mRNA and protein

levels of *c-Myc* and the anti-apoptotic *Bcl-2* and *Bcl-X*$_L$ [21]. An altered rhythm of *PER2* mRNA with lower mesor and amplitude in tumour tissue compared to normal colonic tissue was reported in mice with experimentally-induced colorectal carcinoma [22].

The circadian system and regulation mediated by miRNAs reciprocally influence each other. A distinct daily rhythm of miR-219 and miR-132 was observed in the SCN of mice; interestingly, miR-132 was inducible by light [23]. The rhythmic expression of miRNAs miR-96 and miR-182 was demonstrated in mouse retina, and weak daily rhythms in the expression of mir-16, mir-20a and mir-141 were observed in rat intestine [24, 25]. The precursor form of hepatocyte specific miR-122 exhibits a rhythmic expression; however, the mature strand does not oscillate due to its long half-life, which indicates circadian regulation at a transcriptional level [26]. The rhythmic mRNA expression of Dicer, an enzyme essential for miRNA biogenesis, was identified in the SCN, retina, liver and bone marrow of mice [27]. miRNAs also influence the circadian system. The loss of miRNAs in the liver in *Dicer*$^{flox}$ mice affected the cyclic expression of *PER2* and *PER1* mRNA and protein products, with a significant increase in the mesor and amplitude, while these changes were not observed in the expression of other studied clock genes [28]. Similarly, the miR-192/194 cluster was found to inhibit the expression of *PER2* [29]. In addition, miR-34a was rhythmically expressed in H69 and cholangiocarcinoma cells, and *PER1* was shown to be targeted by this miRNA [30].

Current study is focused on miRNAs that were deregulated in CRC and had a functional connection with the circadian system because of the possible synergic influence of these two factors on tumour progression. miRNAs involved in CRC progression were identified by macroarray analysis, and candidate miRNAs were investigated for an association with the clock gene *PER2*. As such, *PER2* was used as a functional gate because of its involvement in the regulation of oncological malformations and the generation of circadian rhythms.

## Materials and methods

The study included 64 adult patients with CRC. Characteristics of the cohort are provided in Table 1. All patients were exposed to standard hospital practice, with lights on from 6:00 a.m. to 9:00 p.m. The recruitment of the patients was conducted in the First Surgery Department of the Faculty of Medicine of Comenius University and the University Hospital Bratislava in Slovakia from September 2008 to May 2014. All patients were adults, Caucasian and Slovakian residents. Since samples were collected over several years, the cohort could be considered representative of a larger population. Participants were recruited from regular pre-surgery patients by the medical doctors that were co-authors of this study. The experimental protocol, which was in accordance with the Helsinki declaration and approved by the Ethics Committee of Comenius University in Bratislava, was explained to each patient, from whom written informed consent was obtained. Histopathological examinations were performed by a hospital pathologist, and only patients with histologically confirmed colorectal carcinoma were included in the study. During surgery, tissue samples were collected from the primary tumour, as well as from the resected colon $\geq$10 cm proximal and $\geq$2 cm distal of the tumour, in tissues considered to be healthy. Surgeries were conducted between 10:00 a.m. and 1:00 p.m. Tissue samples were collected in liquid nitrogen and then stored at -80°C until further processing. Large RNA and miRNA were extracted from 70 mg tissue samples using RNAzol, according to the manufacturer's instructions (MRC, Inc., USA; Protocol for isolation of Large RNA and small RNA fractions).

### miRNA profiling

To perform miRNA profiling, cDNA was synthesised from 2 μg of miRNA using a miScript II RT kit (Qiagen, Germany). For the identification of altered miRNAs, cDNA from the tumour

**Table 1. Gender, age, tumour location and clinicopathological characteristics of patients.**

| All patients | | n = 64 | % |
|---|---|---|---|
| **Gender** | | | |
| | Male | 38 | 59.4 |
| | Female | 26 | 40.6 |
| **Age** | | | |
| | Mean ± SEM (years) Range (years) | 69 ± 1.4 37–86 | |
| **Tumour location** | | | |
| | Right-side | 25 | 39.1 |
| | Left-side | 39 | 60.9 |
| **Grading stage** | | | |
| | G1 | 11 | 17.2 |
| | G2 | 49 | 76.6 |
| | G3 | 4 | 6.3 |
| **Clinical stage** | | | |
| | I | 4 | 6.3 |
| | IIA, IIB | 29 | 45.3 |
| | IIIA, IIIB | 15 | 23.4 |
| | IVA, IVB | 16 | 25.0 |
| **TNM classification** | | | |
| **Primary tumour invasion** | | | |
| | T1-T2 | 4 | 6.3 |
| | T3 | 48 | 75.0 |
| | T4 | 12 | 18.8 |
| **Regional lymph node** | | | |
| | N0 | 35 | 54.7 |
| | N1 | 13 | 20.3 |
| | N2 | 16 | 25.0 |
| **Distant metastasis** | | | |
| | M0 | 48 | 75.0 |
| | M1 | 16 | 25.0 |

Right-side, anatomic location from caecum to transverse colon; Left-sided, anatomic location from descending colon to rectum; n, number of patients; G, tumour grading; T, tumour invasion; N, nodal status; M, distant metastasis; SEM, standard error of the mean.

was pooled from eight patients. Results were compared to pooled samples of adjacent tissue. Patients of both genders, representing the entire scale of clinicopathological features, were selected for screening. Profiling was performed using the miScript miRNA PCR Array Human Cancer Pathway Finder, according to the manufacturer's instructions (Qiagen, Germany).

### *In silico* analysis

For further analysis, we identified miRNAs with a connection to the circadian system. For this purpose, we employed the TargetScan database [31], in which we searched for miRNA that had predicted interactions with clock gene *PER2* (S1 Table).

### Real-time PCR

To perform real-time PCR, 1 μg of miRNA was polyadenylated using a Poly(A) Tailing kit (Life Technologies, USA). cDNA was then synthesised from 100 ng of polyadenylated

template, using the ImProm-II™ Reverse Transcription kit (Promega, USA) and a primer with a universal tag of the following sequence, 5´-CAGGTCCAGTTTTTTTTTTTTTTTVN-3´ [32].

The detection of relative miRNA expression after profiling was performed by real-time PCR, using the miScript SYBR Green PCR kit (Qiagen, Germany) and the StepOnePlus™ Real-Time PCR System (Applied Biosystems, USA). Data were analysed by StepOne Software v 2.3. We performed arbitrary quantification where the standard curve was specific for a particular gene. Threshold default was 10 standard deviations above the mean fluorescence generated during baseline cycles. A default range of 3–15 cycles was used to establish the baseline and to calculate the Ct values for each sample in the run.

Specific primers used in the assays were: hsa-miR-34a-5p sense, 5´-GCAGTGGCAGTGTCT TAG-3´;antisense, 5´-GGTCCAGTTTTTTTTTTTTTTTTACAAC-3´; and U6 (NR_004394.1) sense, 5´-GCTTCGGCAGCACATATACTAA-3´; antisense, 5´-AAAATATGGAA-CGCTTCA CGA-3´. Details of *PER2* measurements were as previously described [18]. Temperature cycles were 95˚C for 15 min, and then 45 cycles of 55˚C for 30 sec and 70˚C for 30 sec. Expression of small nuclear RNA (snRNA) *U6* was used for the normalisation of miRNA expression data. The specificity of the PCR reactions was validated by melting curve analysis and by sequencing.

## Statistical analysis

The Mann–Whitney *U* test was used for the determination of expression level differences between tumour tissue and its adjacent tissue, as well as between the two groups. Expression values for the adjacent tissue were calculated as the average expression in the proximal and distal tissues. Linear regression analysis was used to determine the correlation between the expression of miRNAs and clock genes. A Kaplan–Meier survival curve and a log-rank test were used to evaluate the overall survival of patients in relation to the median miRNA expression. The starting point for the log-rank test was the day of the surgery. In the histograms, data were provided as a mean ± standard error of the mean (SEM). The level of significance was set at $P < 0.05$.

## Results

Using miRNA macroarray profiling, we identified 17 miRNAs that showed a >100 times expression level in tumour tissue compared to adjacent tissue (Fig 1). From the TargetScan database for predicting miRNA targets, we selected hsa-miR-34a-5p for further analysis, because *in silico* analysis indicated interaction with clock *PER2* (S1 Table).

The expression of miR-34a was significantly up-regulated in tumour tissue compared to matched adjacent tissues for the entire cohort (Mann-Whitney *U* test, $P < 0.001$; Fig 2A), as well in males and females separately (Mann-Whitney *U* test, $P < 0.05$; S1 Fig). For the entire cohort, Kaplan-Meier analysis showed significantly better overall survival for CRC patients with high miR-34a expression in tumour tissue compared to those showing low miR-34a expression (log-rank test, $P = 0.039$; Fig 2B). High and low levels of miR-34a expression were selected on the basis of the median.

After splitting the cohort according to gender, we observed a significant down-regulation of miR-34a in the tumour tissue of male patients with lymph node metastases compared to male patients without lymph node metastases (Mann-Whitney *U* test; $P < 0.05$; Fig 3A). The same trend was preserved in male patients with distant metastases compared to patients without distant metastases (Mann-Whitney *U* test; $P = 0.146$; Fig 3A). These differences were not observed in female patients.

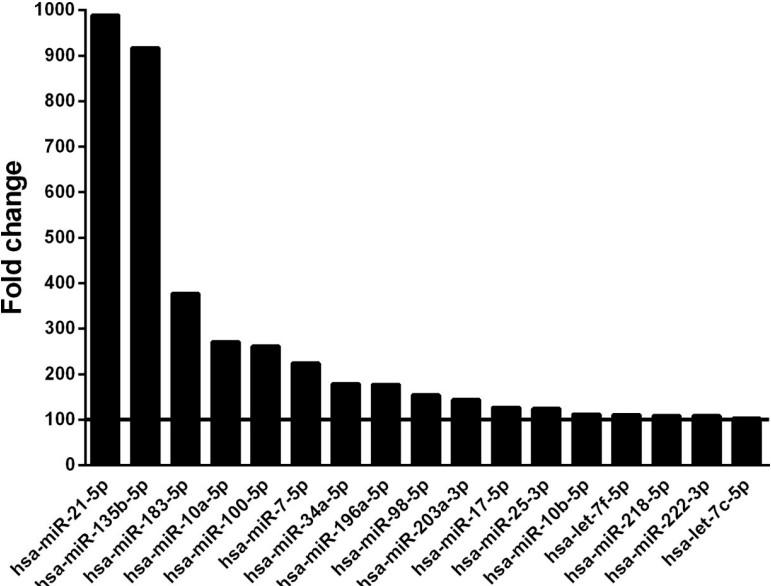

**Fig 1. Fold change of miRNA expression in tumour tissue compared to adjacent proximal tissue, according to macroarray analysis.** The solid line indicates the selection threshold set for expression which was 100 times higher in tumour tissue compared to proximal tissue.

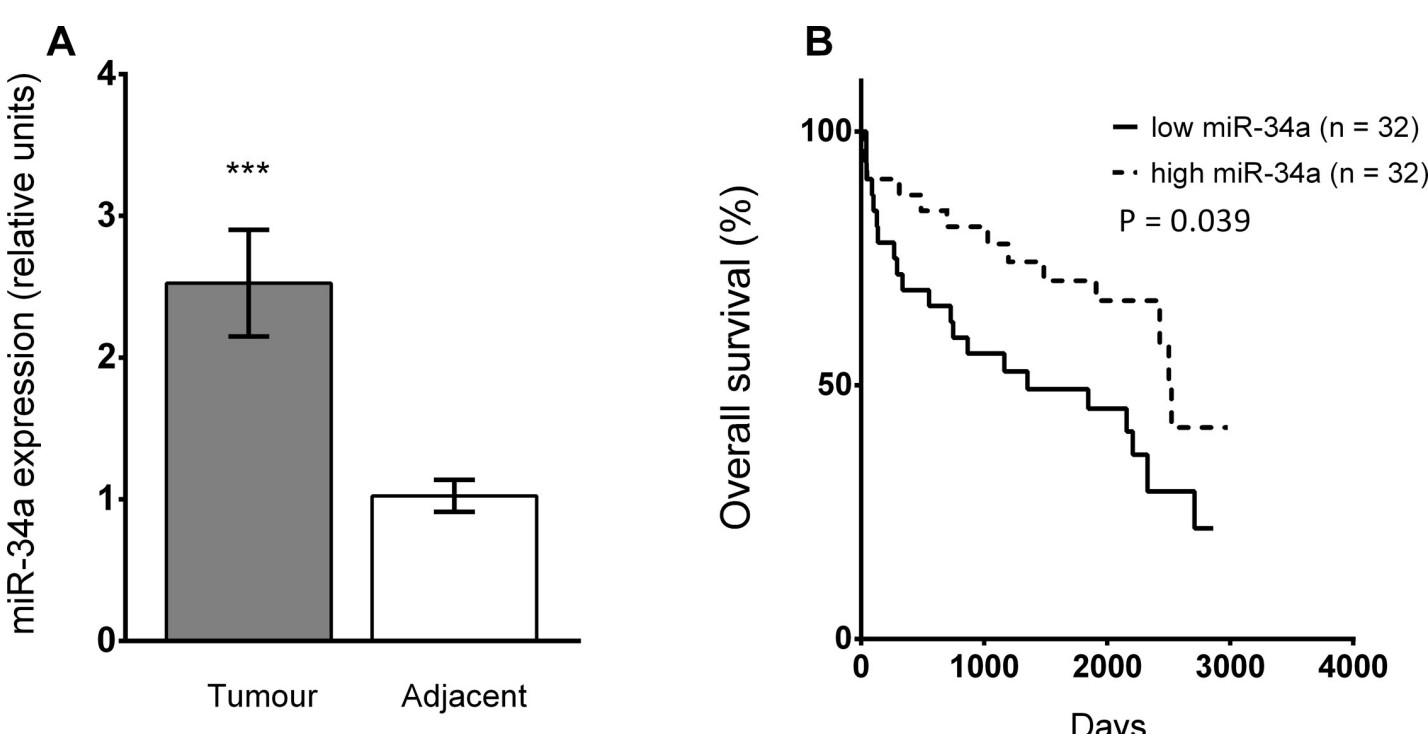

**Fig 2.** Expression of miR-34a-5p (A) in tumour and adjacent tissue in the whole cohort of patients. Data are provided as a mean ± SEM; ***P < 0.001 (n = 64; Mann-Whitney *U* test). (B) Kaplan-Meier survival curve for the entire cohort of patients with low miR-34a expression (≤ median, solid line) and high miR-34a expression (> median, dotted line) in tumour tissue. P = level of significance (log-rank test).

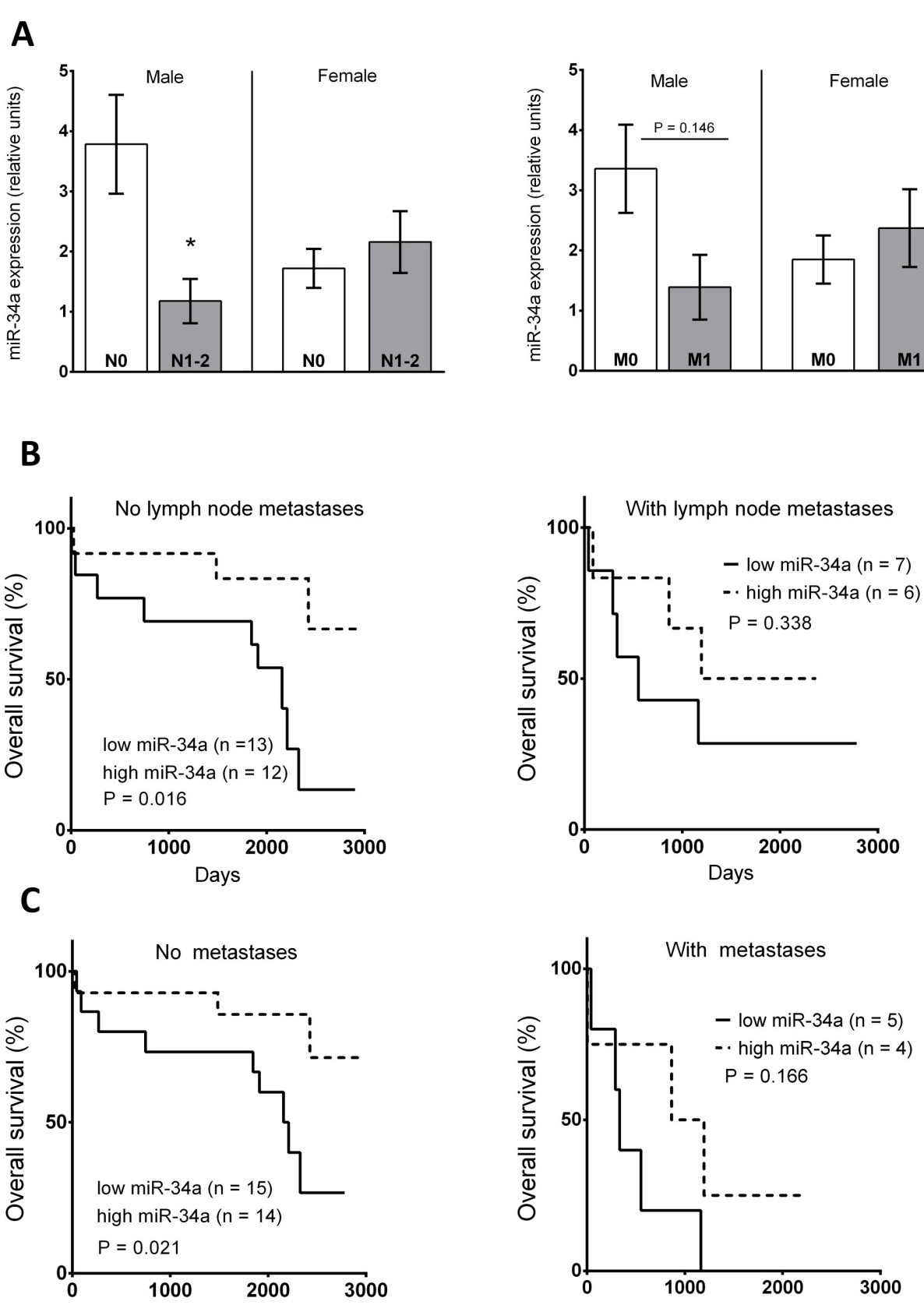

**Fig 3.** The expression of miR-34a (A) in tumour tissue of female (n = 10) and male (n = 25) patients without (N0) and female (n = 16) and male (n = 13) patients with lymph node metastases (N1-2), and female (n = 19) and male (n = 29) patients without (M0) and female (n = 9) and male patients (n = 7) with distant metastases (M1). Data are provided as a mean ± SEM; *p < 0.05; Mann-Whitney U test. (B) Kaplan-Meier survival curve for male patients without/with lymph node metastases according to expression of miR-34a in tumour tissue. (C) Kaplan-Meier survival curve of male patients without/with distant metastasis according to expression of miR-34a in tumour tissue. The solid line indicates low miR-34a expression ($\leq$ median) and the dotted line indicates high miR-34a expression (> median). P = level of significance (log-rank test).

The association between miR-34a expression and patient survival was also analysed in relationship to lymph node involvement and the occurrence of distant metastases. We observed a significant correlation between high miR-34a expression in the tumour tissue of males without lymph node involvement (log rank test, P = 0.016; Fig 3B) and better survival. High expression of miR-34a in male patients without distant metastases was also associated with better survival (log rank test; P = 0.021; Fig 3C). The expression of miR-34a did not significantly predict male survival in the more advanced stages (Fig 3B and 3C). The dependence of survival on cancer progression described above was not observed in females (S2 Fig).

miR-34a expression in tumour tissue correlated with the expression of its predicted target gene *PER2*, which was analysed previously [18]. A significant negative correlation was observed in the specific cohort of patients with cancer at clinical stage IIb and above, with metastases in a maximum of six lymph nodes (regression analysis, R = 0.426; y = -0,06762x + 1079; P = 0.043; Fig 4A), and in patients with T3 stage tumour invasion (tumour invaded through the *muscularis propria* into the pericolorectal tissues) and clinical stage III-IV (regression analysis; R = 0.450; y = -0,05630x + 1118; P = 0.035; Fig 4B).

## Discussion

Macroarray screening identified miR-34a as an up-regulated miRNA in tumour tissue compared to adjacent proximal healthy tissue. Validation by real-time PCR confirmed the increase of miR-34a expression in tumour tissue compared to adjacent tissue for the entire cohort of patients. The expression of miR-34a was associated with patient survival.

miR-34a belongs to the miR-34 miRNA family (miR-34a, miR-34b, miR-34c), and shows robust expression in the brain, lungs and gastrointestinal tract of mice [33]. In humans, its chromosome location is 1p36, which is frequently deleted in malignant tissues, including colorectal tumours [34]. Recently, 77 validated targets of miR-34 family members were identified in various tissues. Most of them promote tumour development, e.g., cell cycle activators (c-Myc and CDK4/6), anti-apoptotic factor (Bcl2), activators of invasion (c-Met, Notch1 and Fra-1) and epithelial-mesenchymal transition inducing factor (SNAIL) [35]. miR-34a expression is promptly activated by DNA damage. It was shown that exposure of HCT116 cells to the DNA intercalating agent Adriamycin causes significant up-regulation of miR-34a. Up-regulation of miR-34a in mice spleen was also observed after cranial exposure to X-rays in females and males [36]. Transient over-expression of miR-34a in human colon cancer cell lines HCT116 and RKO inhibited proliferation and induced senescence-like phenotypes, and suppression of the transcription factor E2F1 [37]. These data supported a tumour suppressive role for miR-34a.

In the current study, the association between higher miR-34a expression in tumour tissue and better overall survival in the entire cohort of patients was consistent with previous data indicating the tumour suppressive properties of miR-34a. Patients with cancer at stages II and III with low miR-34a expression in the tumour showed a higher risk of recurrence than those with high miR-34a expression [38]. Similarly, patients with high miR-34a expression in the tumour had better survival rates than those with low miR-34a expression [39]. High miR-34a expression has only rarely been reported to be associated with worse survival [40].

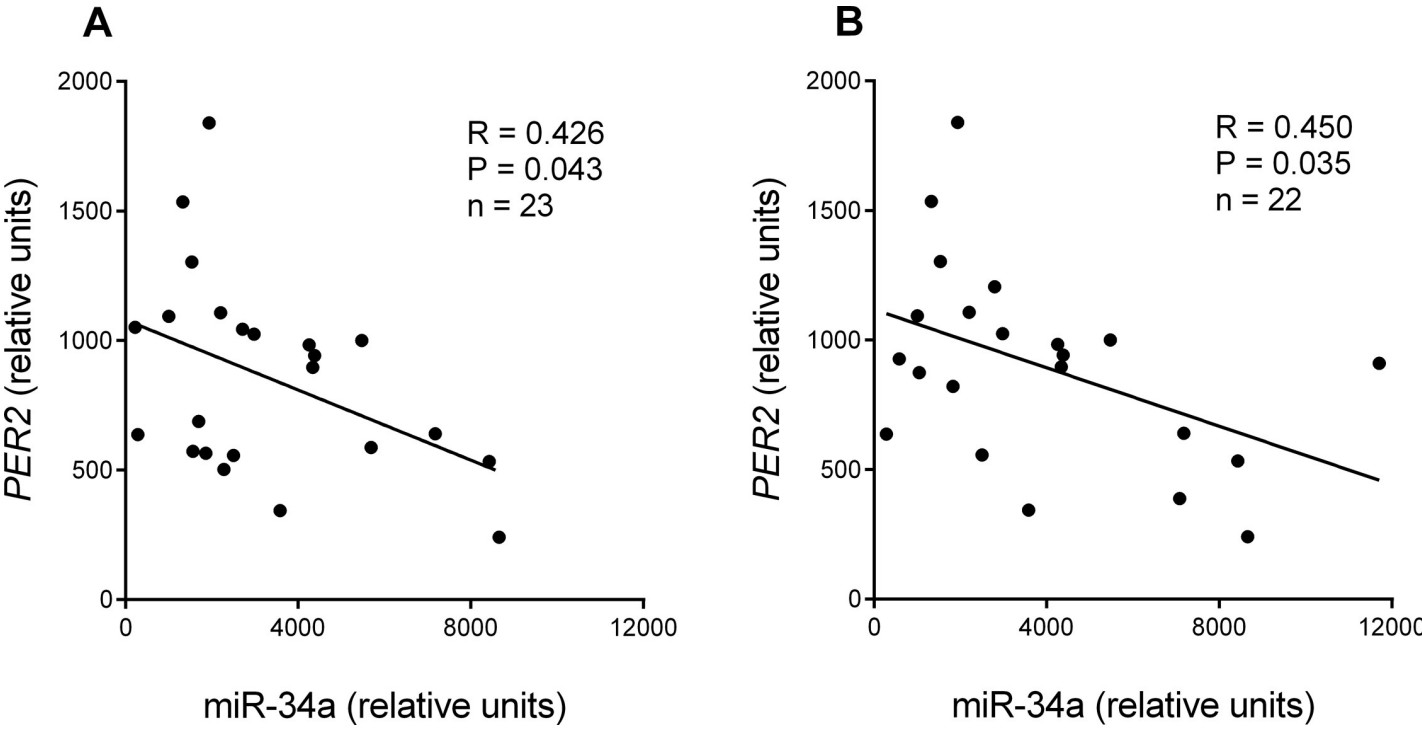

**Fig 4.** Regression analysis of miR-34a and *PER2* expression: (A) in tumour tissue of patients with clinical stage IIb and above, with metastases in a maximum of six lymph nodes; and (B) patients with tumour invasion T3 (tumour invaded through the *muscularis propria* into the pericolorectal tissues) and clinical stage III-IV. Solid line represents significant correlation. R, correlation coefficient; P, level of significance (regression analysis).

One way in which miR-34a could improve survival results from its interaction with tumour suppressor *TP53*. Members of the miR-34 family are targets of *TP53*. p53 increases the expression of miR-34 through the p53-binding region located in the miR-34 promoter [41]. One of the targets of miR-34a is SIRT1 mRNA, which in turn regulates p53-dependent apoptosis by deacetylation and the inactivation of p53. Therefore, there is a positive feedback loop in which p53 induces miR-34a, which conversely down-regulates SIRT1. In this way, miR-34 indirectly increases p53 activity [42]. Interestingly, the induction of miR-34a after DNA damage was also detected in a p53-independent manner. Salzman et al. [43] noticed that there is a pool of mature miR-34a in the cell that remains inactive; however, after irradiation, miR-34 is immediately activated through 5´-end phosphorylation in an ataxia telangiectasia mutated (ATM) and RNA kinase Clp1-dependent manner, enabling association of miR-34a with Argonaute 2 (Ago2) [43]. Another way in which the miR-34 family contributes to tumour suppressive function is that miR-34a, miR-34b*, and miR-34c-5p inhibit the Wnt pathway by repression of Wnt ligand co-receptors (WNT1/3; LRP6), with HMG-box transcription factor (LEF1) and β-catenin. The loss of p53/miR-34 function increases Wnt activity and triggers a tissue-invasive phenotype in CRC cells [44]. In the current study, we did not observe an association between miR-34a expression and survival in either males or females with lymph node or distant metastases, indicating that miR-34a seemed to be a better survival prediction factor in the earlier stages of cancer development.

*In silico* analysis indicated that miR-34a interacted with *PER2* expression. A negative correlation between miR-34a expression and *PER2* expression in tumour tissue was confirmed after cohort clustering. We observed a significant negative correlation in patients with cancer at clinical stage IIb and above, with metastases in a maximum of six lymph nodes, and in patients

with T3 stage tumour invasion (tumour invaded through the *muscularis propria* into the peri-colorectal tissues) and clinical stages III-IV. This relationship has not reported previously been reported, although miR-34a has been shown to inhibit *PER1* expression via the miR-34a conserved regulatory site [30]. It was also shown that *PER2* expression inhibits cell cycle progression [20, 21], and is down-regulated in CRC, with a trend towards a more pronounced decrease with increasing grading stage [18, 45–47]. We propose that miR-34a regulation of cancer progression could be partly explained via the *PER2* gene as well. A negative correlation of *PER2* and miR-34a was present only in patients with more advanced stages of CRC, which was consistent with miR-34a being preferentially associated with survival in patients at earlier cancer stages.

After splitting the cohort according to gender and clinicopathological features, we found that male patients with no lymph node involvement and without distant metastases showed a correlation between higher miR-34a expression in tumour tissue and better overall survival. This association was not observed in females. Sex-dependent differences in survival analyses can be related to the hormonal status of patients; however, the exact mechanism needs to be elucidated. Estradiol was found to down-regulate the expression of miR-34a in the oestrogen receptor α-positive breast cancer cell line-MCF-7 [48]. In our study, although there was trend towards lower expression of miR-34a in females compared to males, it was not statistically significant (S1 Fig). Since miR-34a was similarly expressed in colorectal tumour tissues in males and females, we proposed that its protective role might be mediated by some of its target molecules which showed gender-specific expression.

The expression of miR-34a was up-regulated in tumour tissue compared to adjacent tissue in the whole cohort, with this difference persisted after splitting the cohort according to gender. Previous reports on miR-34a expression in colorectal tumour tissue are inconsistent. The expression of miR-34a in colorectal tumours has been found to be down-regulated [38], up-regulated [49, 50], up-regulated as well as down-regulated [37], or dictated by the clinicopathological stage of patients, as it was found to be increased in higher grading stages [40]. The inconsistency in available data might result from differences in cohort clustering according to patient medical conditions and genders. In our cohort, miR-34a expression was decreased in male patients, but not female patients, with distant metastases and/or lymph node involvement. The gender-dependent feature of this observation would need further evaluation; however, the observed decrease in miR-34a expression could be explained by p53 signalling via the p53-binding region located in the miR-34 promoter. Meta-analysis of *TP53* mutation status in CRC patients showed that around 29% of patients had a type of *TP53* mutation connected with transcriptional inactivity, with patients that had tumours with vascular and lymphatic invasion showing higher frequencies of inactivating *TP53* mutations [51]. However, miR-34a is regulated in a complex manner and other associated pathways have also been described.

Double-negative regulation between the transcription factor SNAIL, which triggers epithelial-mesenchymal transition, and miR-34a was found under *in vitro* conditions. miR-34a represses the expression of *SNAIL*, while SNAIL binds to the promoter of miR-34a and inhibits its transcription, forming a negative feedback loop [52]. A double-negative feedback loop was also found in the regulation of miR-34a through IL-6, which activates the oncogenic transcription factor STAT3, causing the direct repression of miR-34a. miR-34a inhibits the IL-6 receptor and stops IL-6–dependent STAT3 activation [53].

The regulation of miR-34a has also been studied in an epigenetic manner. The entire family of miR-34 is down-regulated by CpG hypermethylation of the miR-34a promoter, which has been observed in CRC but not in normal colonic mucosa [54, 55]. Notably, promoter methylation of miR-34a was higher in CRC tissue from patients with distant metastases and lymph node metastases, compared to those without these systemic manifestations [56]. This was

consistent with the significant down-regulation of tumour miR-34a expression observed in the current study for male patients with lymph node metastases, compared to those without lymph node involvement. These data suggested that epigenetic silencing, which was more pronounced in CRC patients at a more advanced stage, might also contribute to the down-regulation observed in our study. Finally, Wang et al. [49] considered that miR-34a might act in regulatory circuits with positive and negative feedback loops, instead of just mediating unidirectional pathways in the carcinogenesis of CRC.

In conclusion, this study showed that the association of miR-34a with survival that was present in the entire cohort exerted sex-dependent features and differed according to clinico-pathological conditions. Survival was more closely correlates with miR-34a in male patients at earlier cancer stages without distant metastases and/or node involvement, compared to males with CRC at more advanced stages. This was accompanied by a decrease in miR-34a expression in tumours of male patients with lymph node involvement and distant metastases, compared to those with earlier stages of cancer. This profile was not observed in females. Therefore, we propose that miR-34a in earlier cancer stages is regulated in a sex-dependent manner and that a protective role of miR-34a also involves at least one pathway with gender-specific regulation. Besides the well-known tumour suppressive role of miR-34a, we also reported a negative correlation between the expression of the clock gene *PER2* and miR-34a. *PER2* is believed to exert a tumour suppressive role; therefore, in this respect, miR-34a does not seem to play a beneficial role for patients. However, this correlation was only observed in patients with advanced cancer stages, where miR-34a association with survival was weaker. Therefore, we propose that the expression of miR-34a could be considered a better prognostic marker in earlier cancer stages in male patients, rather than female patients.

## Supporting information

**S1 Table. Results from *in silico* analysis with the TargetScan database http://www.targetscan.org/vert_72/.**
(DOCX)

**S1 Fig. Expression of miR-34a in tumour and adjacent tissue in male and female patients.** Data are provided as a mean ± SEM, * indicates an up-regulation in male tumour tissue compared to adjacent tissue ($P < 0.05$; n = 38; Mann-Whitney *U* test) ** indicate an up-regulation in female tumour tissue compared to adjacent tissue ($P < 0.01$; n = 26; Mann-Whitney *U* test).
(TIF)

**S2 Fig. Kaplan-Meier survival curves for female patients according to expression of miR-34a in tumour tissue without/with lymph node metastases and without/with distant metastases.** The solid line indicates low miR-34a expression ($\leq$ median) and the dotted line indicates high miR-34a expression ($>$ median). P = level of significance (log-rank test).
(TIF)

## Acknowledgments

This study was funded by APVV-14-0318, APVV-16-0209 and VEGA 1/0679/19.

## Author Contributions

**Data curation:** Kristina Hasakova, Richard Reis, Marian Vician.

**Formal analysis:** Kristina Hasakova.

**Funding acquisition:** Michal Zeman, Iveta Herichova.

**Methodology:** Iveta Herichova.

**Resources:** Kristina Hasakova, Iveta Herichova.

**Supervision:** Iveta Herichova.

**Writing – original draft:** Kristina Hasakova.

**Writing – review & editing:** Richard Reis, Marian Vician, Michal Zeman, Iveta Herichova.

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
