## [Decision Letter · Decision Letter 0]

23 Jul 2019

PONE-D-19-15253

Expression of miR-34a-5p is up-regulated in human colorectal cancer and correlates with survival and clock gene PER2 expression

PLOS ONE

Dear Dr Herichova,

Thank you for submitting your manuscript to PLOS ONE. After careful consideration, we feel that it has merit but does not fully meet PLOS ONE’s publication criteria as it currently stands. Therefore, we invite you to submit a revised version of the manuscript that addresses the points raised during the review process.

We would appreciate receiving your revised manuscript by Sep 06 2019 11:59PM. To enhance the reproducibility of your results, we recommend that if applicable you deposit your laboratory protocols in protocols.io, where a protocol can be assigned its own identifier (DOI) such that it can be cited independently in the future. For instructions see: http://journals.plos.org/plosone/s/submission-guidelines#loc-laboratory-protocols

We look forward to receiving your revised manuscript.

Kind regards,

Ondrej Slaby

Academic Editor

PLOS ONE

Journal Requirements:

2)  Please provide author names on the title page of your MS as Firstname Lastname. (Currently names are provided as Lastname Firstname, which might cause issues with PubMed on down the line.)

3) In your Methods section, please provide additional information about the participant recruitment method and the demographic details of your participants. Please ensure you have provided sufficient details to replicate the analyses such as: a) the recruitment date range (month and year), b) a description of any inclusion/exclusion criteria that were applied to participant recruitment, c) a statement as to whether your sample can be considered representative of a larger population, d) a description of how participants were recruited, and e) descriptions of where participants were recruited and where the research took place.

4) Please also provide the full name of the Ethics Committee that approved the study, and provide the approval or permit number that was issued when the study was approved.

5) We note that you have included the phrase “data not shown” in your manuscript. Unfortunately, this does not meet our data sharing requirements. PLOS does not permit references to inaccessible data. We require that authors provide all relevant data within the paper, Supporting Information files, or in an acceptable, public repository. Please add a citation to support this phrase or upload the data that corresponds with these findings to a stable repository (such as Figshare or Dryad) and provide and URLs, DOIs, or accession numbers that may be used to access these data. Or, if the data are not a core part of the research being presented in your study, we ask that you remove the phrase that refers to these data.

<h1>****</h1>

Reviewers' comments:

Reviewer's Responses to Questions

**Comments to the Author**

1. Is the manuscript technically sound, and do the data support the conclusions?

Reviewer #1: Partly

Reviewer #2: Yes

2. Has the statistical analysis been performed appropriately and rigorously? 

Reviewer #1: No

Reviewer #2: Yes

3. Have the authors made all data underlying the findings in their manuscript fully available?

Reviewer #1: Yes

Reviewer #2: Yes

4. Is the manuscript presented in an intelligible fashion and written in standard English?

Reviewer #1: No

Reviewer #2: Yes

5. Review Comments to the Author

Reviewer #1: The authors describe an interesting idea of circadian system regulation of miRNA expression in colorectal carcinoma. While the idea is very interesting, I have reservation to both the study design and the manuscript.

The authors need to proof read their manuscript, not only concerning usage of English per se but also scientific language (tumor, cancer when describing colorectal carcinoma - and then they should stick to that term). The nomenclature of genes is somewhat confusing, sometimes small italics, sometimes large italics when describing human genes.

Line 68 - when describing studies on CRC and miRNA, only 1 reference is used, there are many studies.....

line 69 - description of circadian system is unclear

line 70 - oncologic processes? what do you mean by that?

line 86 - please change p53 to TP53 if describing a gene (unclear)

Methods - please include a sentence if the ethics committee approval is in accordance with the Helsinki declaration

table 1 - please show age of patients at diagnosis as range

location of tumor - please explain left and right tumor location

grading and staging systems - please use the current nomenclature of grading system G1-G4

methods - i am unclear about the profiling - which pts were actually chosen for the microarray? how did you measure 2 ug of miRNA? i would think that it was 2 ug of total RNA...

why were the tumor samples and adjacent samples pooled for the profiling? this part is unclear. hy only 8 patients? how many arrays did the authors actually use?

real-time PCR - again, how did you isolate miRNA? or did you use total RNA with miRNA? what is arbitrary quantification? why did you use it?

the figures are missing the number of patients that were actually included in the analysis. CRC is a common disease, while the results are interesting, the number of patients is very low and should be increased.

Moreover, the authors failed to show any functional interactions between miR34a and the clock gene PER2 which should be included in this study ...the authors claim they showed this in the abstract but only a negative corelation was shown.

Reviewer #2: The paper does not include a proper introduction into the topic. A more detailed information on the mechanism of clock genes contribution to cancer should be given (not just a single reference).

- in the methods section it is not fully clear whether samples from "STAGE IV" patients were originating from primary carcinoma tissue or from metastatic tissue. In case of the former it would only be applicable to patients with synchronous metastases and as such it should be clearly described in the text.

6. PLOS authors have the option to publish the peer review history of their article (what does this mean?). If published, this will include your full peer review and any attached files.

Reviewer #1: No

Reviewer #2: No

---

## [Author Response · Author response to Decision Letter 0]

4 Sep 2019

Editorial comments:

We checked and corrected the manuscript according to PLOS ONE's style requirements.

2) Please provide author names on the title page of your MS as Firstname Lastname. (Currently names are provided as Lastname Firstname, which might cause issues with PubMed on down the line.)

Thank you for the comment, in revised MS we provided the authors names as first name and last name.

3) In your Methods section, please provide additional information about the participant recruitment method and the demographic details of your participants. Please ensure you have provided sufficient details to replicate the analyses such as: a) the recruitment date range (month and year), b) a description of any inclusion/exclusion criteria that were applied to participant recruitment, c) a statement as to whether your sample can be considered representative of a larger population, d) a description of how participants were recruited, and e) descriptions of where participants were recruited and where the research took place.

Additional information about participant recruitment method, exclusion criteria and demographic details was added into the method section.

We included text:

“The recruitment of the patients was conducted in the First Surgery Department of the Faculty of Medicine of Comenius University and the University Hospital Bratislava in Slovakia from September 2008 to May 2014. All patients were adults, Caucasian and Slovakian residents. Since samples were collected over several years, the cohort could be considered representative of a larger population. Participants were recruited from regular pre-surgery patients by the medical doctors that were co-authors of this study. The experimental protocol, which was in accordance with the Helsinki declaration and approved by the Ethics Committee of Comenius University in Bratislava, was explained to each patient, from whom written informed consent was obtained. Histopathological examinations were performed by a hospital pathologist, and only patients with histologically confirmed colorectal carcinoma were included in the study.”

4) Please also provide the full name of the Ethics Committee that approved the study, and provide the approval or permit number that was issued when the study was approved.

We provided the full name of the Ethic Committee in the chapter “Material and methods” and approval number.

Ethics Committee of the Comenius University in Bratislava (ECH 19001)

5) We note that you have included the phrase “data not shown” in your manuscript. Unfortunately, this does not meet our data sharing requirements. PLOS does not permit references to inaccessible data. We require that authors provide all relevant data within the paper, Supporting Information files, or in an acceptable, public repository. Please add a citation to support this phrase or upload the data that corresponds with these findings to a stable repository (such as Figshare or Dryad) and provide and URLs, DOIs, or accession numbers that may be used to access these data. Or, if the data are not a core part of the research being presented in your study, we ask that you remove the phrase that refers to these data.

We provided these data as a supplementary figure 2.

Reviewers' comments:

Reviewer's Responses to Questions

Comments to the Author

1. Is the manuscript technically sound, and do the data support the conclusions?

Reviewer #1: Partly

Reviewer #2: Yes

2. Has the statistical analysis been performed appropriately and rigorously? 

Reviewer #1: No

Reviewer #2: Yes

3. Have the authors made all data underlying the findings in their manuscript fully available?

Reviewer #1: Yes

Reviewer #2: Yes

4. Is the manuscript presented in an intelligible fashion and written in standard English?

Reviewer #1: No

Reviewer #2: Yes

5. Review Comments to the Author

Reviewer #1: 

The authors describe an interesting idea of circadian system regulation of miRNA expression in colorectal carcinoma. While the idea is very interesting, I have reservation to both the study design and the manuscript.

The authors need to proof read their manuscript, not only concerning usage of English per se but also scientific language (tumor, cancer when describing colorectal carcinoma - and then they should stick to that term). 

Thank you for the comment, the manuscript was proofread and corrected by authors and also by professional agency (please, see attached certificate). We avoided the term “cancer tissue” which was replaced by “tumour tissue” in the whole manuscript.

The nomenclature of genes is somewhat confusing, sometimes small italics, sometimes large italics when describing human genes.

The manuscript was proofread and the nomenclature of genes were united according the web page:

https://www.biosciencewriters.com/Guidelines-for-Formatting-Gene-and-Protein-Names.aspx

paragraph:

Humans, non-human primates, chickens, and domestic species: Gene symbols contain three to six italicized characters that are all in upper-case (e.g., AFP). Gene symbols may be a combination of letters and Arabic numerals (e.g., 1, 2, 3), but should always begin with a letter; they generally do not contain Roman numerals (e.g., I, II, III), Greek letters (e.g., α, β, γ), or punctuation. Protein symbols are identical to their corresponding gene symbols except that they are not italicized (e.g., AFP).

Line 68 - when describing studies on CRC and miRNA, only 1 reference is used, there are many studies.....

We added more studies referring about changes of miRNA in CRC and included text:

„In tumours of colorectal carcinoma up-regulated miRNAs include miR-17, miR-155, miR-146, miR-221, miR-31, miR196, miR-20a, miR92a, miR-130b, miR181b, miR-183, miR-196, miR-203, miR-215, miR-224 and miR-25, while down-regulated miRNAs include miR-10, miR-30, miR-22, miR-93, miR-126, miR-132, miR-139, miR-15, miR-195, miR340 and miR-106a [3, 10, 11].”

line 69 - description of circadian system is unclear

The chapter “Introduction” describing the circadian system was completely rewritten.

line 70 - oncologic processes? what do you mean by that?

The word “oncologic processes” was omitted from the manuscript. The sentence was replaced. Chapter “Introduction” was rewritten.

line 86 - please change p53 to TP53 if describing a gene (unclear)

Thank you for your comment, nomenclature of gene TP53 was corrected in the whole manuscript.

Methods - please include a sentence if the ethics committee approval is in accordance with the Helsinki declaration

Thank you for the comment, the sentence claiming that ethics committee approval was in accordance with the Helsinki declaration was added in the chapter “Material and methods”.

table 1 - please show age of patients at diagnosis as range

Thank you for the comment. Patient age at the time of surgery was added into Table 1. Range was 37 - 86 years.

location of tumor - please explain left and right tumor location

Explanation of the right and left-sided tumour location was added to the Table 1 legend.

grading and staging systems - please use the current nomenclature of grading system G1-G4

Thank you for the suggestion, the nomenclature of grading system was changed in the Table 1.

methods - i am unclear about the profiling - which pts were actually chosen for the microarray? how did you measure 2 ug of miRNA? i would think that it was 2 ug of total RNA...

why were the tumor samples and adjacent samples pooled for the profiling? this part is unclear. hy only 8 patients? how many arrays did the authors actually use?

real-time PCR - again, how did you isolate miRNA? or did you use total RNA with miRNA? what is arbitrary quantification? why did you use it?

For the profiling purposes we selected eight patients of both gender that represent the entire scale of clinicopathological features of our cohort. We could not use of cDNA from the whole cohort because we needed only 40ul cDNA for the whole array and cohort consisted from 64 patients. Pipetting of too low volumes (less than 1µl) is not exact, therefore we selected representative part of cohort and validated data with the use of whole sample set (64 samples). 

We used two arrays, one for tumour tissue and one for the proximal adjacent tissue. By comparison of results from these two arrays we selected miR-34a-5p as a candidate miRNA to be estimated in the whole cohort.

Arbitrary units are the most frequent way how to interpret gene expression data. We prepared calibration curve from logarithmically diluted sample with high expression that was used to calculate relative expression. 

The concentration of miRNA was measured by spectrophotometry after miRNA isolation from tissue and its dilution in redistill water. 2µg of miRNA was calculated as a corresponding volume from a diluted sample. We performed isolation in two steps. We isolated large RNA (mRNA and ribosomal RNA) and small RNA molecules separately using RNAzol according to the manufacturer’s instructions (MRC, USA, Protocol for isolation of Large RNA and small RNA fractions). We did not use total RNA containing miRNA. https://www.mrcgene.com/wp-content/uploads/2017/04/RNAzolRTMarch2017.pdf

This information is provided in chapter “Material and methods”.

the figures are missing the number of patients that were actually included in the analysis.

Thank you for the comment. The number of patients were added into the figures or figure legends.

CRC is a common disease, while the results are interesting, the number of patients is very low and should be increased.

The sample size of our study (n=64) is comparable with other studies focused on the same topic published by European authors.

Mazzoccoli G, Panza A, Valvano MR, Palumbo O, Carella M, Pazienza V, Biscaglia G, Tavano F, Di Sebastiano P, Andriulli A, et al. 2011. Clock gene expression levels and relationship with clinical and pathological features in colorectal cancer patients. Chronobiol Int. 28(10):841–851. doi: 10.3109/07420528.2011.615182; cited by 67 papers, 19 patients in cohort

Mazzoccoli G, Colangelo T, Panza A, Rubino R, De Cata A, Tiberio C, Valvano MR, Pazienza V, Merla G, Augello B, et al. 2016. Deregulated expression of cryptochrome genes in human colorectal cancer. Mol Cancer. 15:6. doi: 10.1186/s12943-016-0492-8; cited by 12 papers , 50 patients in cohort

Karantanos T, Theodoropoulos G, Gazouli M, Vaiopoulou A, Karantanou C, Lymberi M, Pektasides D. 2013. Expression of clock genes in patients with colorectal cancer. Int J Biol Markers. 28(3):280-5. doi: 10.5301/jbm.5000033; cited by 29 papers, 42 patients in cohort

Krugluger W, Brandstaetter A, Kállay E, Schueller J, Krexner E, Kriwanek S, Bonner E, Cross HS. 2007. Regulation of genes of the circadian clock in human colon cancer: reduced period-1 and dihydropyrimidine dehydrogenase transcription correlates in high-grade tumors. Cancer Res. 67(16):7917-22; cited by 72 papers, 30 patients in cohort

Momma T, Okayama H, Saitou M, Sugeno H, Yoshimoto N, Takebayashi Y, Ohki S, Takenoshita S. 2017. Expression of circadian clock genes in human colorectal adenoma and carcinoma. Oncol Lett. 14(5):5319-5325. doi: 10.3892/ol.2017.6876; cited by 9 papers, 51 patients in cohort

Huisman SA, Ahmadi AR, IJzermans JN, Verhoef C, van der Horst GT, de Bruin RW. 2016. Disruption of clock gene expression in human colorectal liver metastases. Tumour Biol. 37(10):13973-13981; cited by 6 papers, 15 patienst in cohort

 Mostafaie N, Kállay E, Sauerzapf E, Bonner E, Kriwanek S, Cross HS, Huber KR, Krugluger W. 2009. Correlated downregulation of estrogen receptor beta and the circadian clock gene Per1 in human colorectal cancer. Mol Carcinog. 48(7):642-7. doi: 10.1002/mc.20510. cited by 64 papers, 20 patienst in cohort

Bandrés E, Cubedo E, Agirre X, Malumbres R, Zárate R, Ramirez N, Abajo A, Navarro A, Moreno I, Monzó M, García-Foncillas J. 2006. Identification by Real-time PCR of 13 mature microRNAs differentially expressed in colorectal cancer and non-tumoral tissues. Mol Cancer. 19;5:29. Cited by 684 papers, 12 patients in cohort

Moreover, the authors failed to show any functional interactions between miR34a and the clock gene PER2 which should be included in this study ...the authors claim they showed this in the abstract but only a negative corelation was shown.

Statement that functional interactions between miR-34a and PER2 exists was removed from abstract and replaced by statement: “From in silico analysis, miR-34a-5p was selected as showing a computer-predicted interaction with PER2.”

Reviewer #2: 

The paper does not include a proper introduction into the topic. A more detailed information on the mechanism of clock genes contribution to cancer should be given (not just a single reference).

Thank you for the comment. We rewrote the chapter “Introduction” and described mechanism of clock gene (and particularly PER2) contribution to cancer progression in more detail.

- in the methods section it is not fully clear whether samples from "STAGE IV" patients were originating from primary carcinoma tissue or from metastatic tissue. In case of the former it would only be applicable to patients with synchronous metastases and as such it should be clearly described in the text.

Samples from patients with distant metastases were obtained from the primary tumour tissue. This information was added to the chapter “Materials and methods” (line 146).

---

## [Decision Letter · Decision Letter 1]

14 Oct 2019

Expression of miR-34a-5p is up-regulated in human colorectal cancer and correlates with survival and clock gene PER2 expression

PONE-D-19-15253R1

Dear Dr. Herichova,

We are pleased to inform you that your manuscript has been judged scientifically suitable for publication and will be formally accepted for publication once it complies with all outstanding technical requirements.

With kind regards,

Ondrej Slaby

Academic Editor

PLOS ONE

Additional Editor Comments (optional):

Reviewers' comments:

Reviewer's Responses to Questions

**Comments to the Author**

1. If the authors have adequately addressed your comments raised in a previous round of review and you feel that this manuscript is now acceptable for publication, you may indicate that here to bypass the “Comments to the Author” section, enter your conflict of interest statement in the “Confidential to Editor” section, and submit your "Accept" recommendation.

Reviewer #1: All comments have been addressed

Reviewer #2: All comments have been addressed

2. Is the manuscript technically sound, and do the data support the conclusions?

Reviewer #1: (No Response)

Reviewer #2: Yes

3. Has the statistical analysis been performed appropriately and rigorously? 

Reviewer #1: (No Response)

Reviewer #2: Yes

4. Have the authors made all data underlying the findings in their manuscript fully available?

Reviewer #1: (No Response)

Reviewer #2: Yes

5. Is the manuscript presented in an intelligible fashion and written in standard English?

Reviewer #1: (No Response)

Reviewer #2: Yes

6. Review Comments to the Author

Reviewer #1: The authors have addressed all issues. The manuscript has been greatly improved. However, I believe that discussion should be slightly rewritten - the authors start with a short summary of their study but a more appropriate start would be to discuss CRC and the importance of easily obtainable markers for prognosis of patients.

Reviewer #2: (No Response)

7. PLOS authors have the option to publish the peer review history of their article (what does this mean?). If published, this will include your full peer review and any attached files.

Reviewer #1: No

Reviewer #2: No

---

## [Editor Report · Acceptance letter]

18 Oct 2019

PONE-D-19-15253R1 

Expression of miR-34a-5p is up-regulated in human colorectal cancer and correlates with survival and clock gene *PER2* expression 

Dear Dr. Herichova:

I am pleased to inform you that your manuscript has been deemed suitable for publication in PLOS ONE. Congratulations! Your manuscript is now with our production department. 

With kind regards,

on behalf of

Dr. Ondrej Slaby 

Academic Editor

PLOS ONE